# Equine Genital Squamous Cell Carcinoma Associated with EcPV2 Infection: RANKL Pathway Correlated to Inflammation and Wnt Signaling Activation

**DOI:** 10.3390/biology10030244

**Published:** 2021-03-21

**Authors:** Samanta Mecocci, Ilaria Porcellato, Federico Armando, Luca Mechelli, Chiara Brachelente, Marco Pepe, Rodolfo Gialletti, Benedetta Passeri, Paola Modesto, Alessandro Ghelardi, Katia Cappelli, Elisabetta Razzuoli

**Affiliations:** 1Department of Veterinary Medicine, University of Perugia, Perugia 06123, Italy; samanta.mecocci@studenti.unipg.it (S.M.); ilariaporcellatodvm@gmail.com (I.P.); luca.mechelli@unipg.it (L.M.); chiara.brachelente@unipg.it (C.B.); marco.pepe@unipg.it (M.P.); rodolfo.gialletti@unipg.it (R.G.); 2Department of Veterinary Science, University of Parma, Parma 43126, Italy; benedetta.passeri@unipr.it; 3Department of Pathology, University of Veterinary Medicine Hannover, Hannover 30559, Germany; 4Centro di Ricerca sul Cavallo Sportivo, University of Perugia, Perugia 06123, Italy; 5National Reference Center of Veterinary and Comparative Oncology (CEROVEC), Genova 16132, Italy; paola.modesto@izsto.it (P.M.); elisabetta.razzuoli@izsto.it (E.R.); 6Azienda Usl Toscana Nord-Ovest, UOC Ostetricia e Ginecologia, Ospedale Apuane, Massa 54100, Italy

**Keywords:** Wnt pathway, RANKL pathway, inflammation, *IL17A*, equine, EcPV2, E6 expression, *IL23*, *IL12*

## Abstract

**Simple Summary:**

Equine genital squamous cell carcinomas (egSCCs) associated with papilloma virus (PV) infection have been recently proposed as model for human PV-induced SCC. In both species, PV mucosal infections often induce cervical, oropharyngeal, penile, anal, vaginal, and vulvar cancer. The aim of this study was to clarify the molecular mechanisms behind egSCCs associated with equine papillomavirus 2 (EcPV2) infection investigating receptor activator of nuclear factor-kappa B ligand (RANKL), Wnt, and interleukin (IL)17 signaling pathways. RANKL has been recently demonstrated to play a crucial role in several human tumors, associated with a poor prognosis and metastatic spread; novel targeted therapies through RANKL silencing monoclonal antibodies have been undertaken. EcPV2-*E6* DNA was checked, and viral presence was confirmed in 91% of cases, whereas oncogene expression was 60.8% for *E6* and 34.7% for *E2*. *RANKL*, *NFKBp50*, *NFKBp65*, *IL6*, *IL17*, *IL23p19*, *IL8*, *IL12p35*, *IL12p40*, *BCATN1*, *FOSL1*, and *LEF1* gene expression showed a significant upregulation in tumor samples compared to healthy tissues. Our results describe an inflammatory environment characterized by the increased expression of several cytokines and the activation of RANKL/RANK, IL17A, and canonical and non-canonical Wnt signaling pathways. These results may be helpful to identify new targets for immunotherapy strategies confirming egSCCs as a model for the human disease.

**Abstract:**

Equine genital squamous cell carcinomas (egSCCs) are among the most common equine tumors after sarcoids, severely impairing animal health and welfare. *Equus caballus* papillomavirus type 2 (EcPV2) infection is often related to these tumors. The aim of this study was to clarify the molecular mechanisms behind egSCCs associated with EcPV2 infection, investigating receptor activator of nuclear factor-kappa B ligand (RANKL) signaling in NF-kB pathway, together with the Wnt and IL17 signaling pathways. We analyzed the innate immune response through gene expression evaluation of key cytokines and transcription factors. Moreover, Ki67 index was assessed with immunohistochemistry. EcPV2-*E6* DNA was checked, and viral presence was confirmed in 21 positive out to 23 cases (91%). Oncogene expression was confirmed in 14 cases (60.8%) for *E6* and in 8 (34.7%) for *E2*. *RANKL*, nuclear factor kappa-light-chain-enhancer of activated B cells (*NFKB*)-*p50*, *NFKBp65*, interleukin *(IL)-6*, *IL17*, *IL23p19*, *IL8*, *IL12p35*, *IL12p40*, *β-catenin (BCATN1)*, *FOS like 1 (FOSL1)*, and lymphoid enhancer binding factor 1 *(LEF1)* showed a significant upregulation in tumor samples compared to healthy tissues. Our results describe an inflammatory environment characterized by the activation of RANKL/RANK and IL17 with the relative downstream pathways, and a positive modulation of inflammatory cytokines genes such as *IL6* and *IL8*. Moreover, the increase of *BCATN1*, *FOSL1*, and *LEF1* gene expression suggests an activation of both canonical and non-canonical Wnt signaling pathway that could be critical for carcinogenesis and tumor progression.

## 1. Introduction

Equine genital squamous cell carcinomas (egSCCs) are among the most common neoplasms after sarcoids and the most frequent tumor of the external genitalia in *Equus caballus* species, especially in males [1]. Like for humans, where the association between human papillomavirus (HPV) and cervical carcinoma is now well established, equine papillomaviruses (EcPV) seem to have a predominant role in the pathogenesis of gSCC in horses [2]. Despite numerous studies available on papillomaviruses (PVs), many questions are still unanswered about the role of innate immunity in the early stages of infection and the molecular basis of tissue tropism [3]. In order to fill these gaps, it is useful to identify alternative animal models to those in use (dog, bovine, mouse, rabbit), which over the years have provided important, yet not exhaustive, information. More recently, horses have been suggested as an animal model for human PV infection, because of morphological and molecular similarities [1,4].

Chronic inflammation can lead to cancer development, and most solid tumors are infiltrated by inflammatory cells [5,6]; furthermore, tumor progression and response to treatment are influenced by the interaction between cancer cells and the tumor immune microenvironment (TIME). This information has been reported both for HPV and EcPV2-associated cancer [4,5]. At the same time, tumors can develop several mechanisms leading to immunosuppression and alteration of the immune response, such as the imbalance between Th1/Th2 and Th17/Treg in PV-associated lesions, which could support tumor development, as reported in cervical cancer [4,7].

Recently, the Wnt signaling pathway, a key molecular cascade for cell proliferation, has been associated with a variety of diseases, including cancer, with expression modulation and mutations [8,9]. Interestingly, the canonical Wnt/β-catenin pathway in human cervical cancers seems to be influenced by HPV type 16 that leads to an increased cytoplasmic and nuclear β-catenin expression, accelerating the carcinogenetic process [10,11]. Both Wnt canonical and non-canonical pathways could be involved in the pathogenesis of several human malignancies. Interestingly, some pieces of evidence show that cancer cells use the non-canonical signaling pathway to acquire the ability to migrate and metastasize [12]. Wnt signaling network is also involved in the modulation of the inflammatory response and, as recently reported [13], its activation is an important feature in human penile, oropharyngeal, and cervix squamous cell carcinoma [10,14,15,16]. A recent study demonstrated an increased number of cells expressing Fos-like antigen-1 (fra1), a Fos-group AP-1 transcription factor component, in equine penile squamous cell carcinoma (epSCC). This molecule is not only a direct target of the Wnt signaling pathway but also of the interleukin 17 (IL17) signaling pathway [1]. This protein, together with IL17A, is also involved in another pathway recently associated with HPV genital cSCC progression, the RANK/RANKL pathway. Nowadays, at least seven downstream signaling pathways for RANK have been described [17]. The best-known is the RANKL/RANK/OPG pathway, which has a pivotal role in bone metabolism, involved in the physiological process of bone remodeling, and osteoclast genesis [18]. The binding of RANKL to RANK activates a wide range of signaling cascades, including NF-kB [19].

As suggested by these findings, there is a close relationship among the bone and immune systems, and RANKL is one of the principal molecules that confirms this link. In fact, the RANKL-mediated RANK activation has first been described as a pathway increasing the ability of dendritic cells (DCs) to stimulate naive T cell proliferation and enhance DCs survival [20,21]. Moreover, RANKL signaling is crucial for the development of various immune organs such as the thymus, lymph nodes, and bone marrow, and is also implicated in chronic inflammatory conditions of the intestine, skin, and nervous system [22]. In humans, an increased expression of RANK/RANKL has been recently demonstrated in several tumors. A high percentage of carcinoma cells express RANK mRNA/protein at various levels, especially in breast and prostate cancers, chondrosarcomas, and osteosarcomas [23]. RANK expression has been frequently reported to be predictive of a poor prognosis and is often associated with the metastatic spread of the tumor, affecting cancer cell motility [23]. On the other hand, the *RANKL* expression is associated with tumor-infiltrating immune cells, especially T cells, within the TIME and the locoregional lymph nodes [24]. Thus, these numerous findings strengthen the hypothesis of RANKL/RANK key role in the oncogenesis. Indeed, an anti-tumor strategy may also involve the inhibition of RANKL via monoclonal antibodies, such as denosumab [24,25]. High levels of RANKL/RANK have been demonstrated also in human cervical cancer cells attributing to RANKL signaling a crucial role and paving the way for a targeted therapy through monoclonal antibodies [26]. However, in HPV as in equine PV-related cancer, recently suggested as a model to human HPV-induced SCC [4], studies on these topics are few in number, and much missing data are highly demanded [15].

The aim of this study was to deepen current knowledge on molecular features of egSCCs associated with EcPV2 infection, investigating RANKL signaling in NF-kB pathway together to IL-17 and Wnt signaling. Moreover, the innate immune response framework was investigated through gene expression evaluation of correlated cytokines and transcription factors for a broad vision of the phenomenon.

## 2. Materials and Methods

### 2.1. Samples

Twenty-three cases from the archives of the Veterinary Medicine Department of the Perugia University and the Veterinary Science Department of the Parma University were retrospectively selected and included in the Tumor group (T). Inclusion criteria were histological diagnosis of papilloma, carcinoma in situ, or squamous cell carcinoma; vulvar/vaginal or penile localization of the lesions; availability of >0.5 cm^2^ of formalin-fixed, paraffin-embedded (FFPE) tumor tissue evaluated on section.

A total of 10 normal penile and vulvar samples, obtained from the slaughterhouse, were used as control group (C). Samples were formalin-fixed and paraffin-embedded with the same protocol followed for the tumor tissues. Moreover, histopathological diagnosis and EcPV2 DNA presence were assessed in each sample. Only healthy tissue resulted negative for EcPV2 were included in the study.

### 2.2. Histopathological Diagnosis and Immunohistochemistry

After confirmation of the histological diagnosis, mitotic count was assessed as per standardized method [27]. For immunohistochemistry, FFPE samples were cut into 5 μm sections, mounted on poly-L-lysine-coated slides, dewaxed, and rehydrated. Immunolabeling was obtained with standard protocols on serial sections, with an anti-Ki67 antibody as previously reported [4]. The immunolabeling was revealed with 3-amino-9-ethilcarbazole (Abcam), and Mayer’s hematoxylin was applied as a counterstain. Ki67 index was evaluated on 1000 cells using ImageJ cell counter and expressed as a percentage, as previously reported [28].

### 2.3. DNA Extraction and EcPV2 Detection

Two to three, 5 μm thick sections were cut from each FFPE block (both T and C samples) and subjected to DNA extraction for the evaluation of EcPV2 presence, which was carried out using the AllPrep DNA FFPE Kit (Qiagen, Venlo, the Netherlands) following the manufacturer’s instructions. Previously described primers for EcPV2-*E6* DNA and related specific probes were used for the virus detection [29], and beta-2-microglobulin (*B2M*) gene was targeted to assess DNA amplifiability. Primer sequences are reported in Table 1. Real-time PCR was performed in a CFX96 Real-Time System using Taq DNA Polymerase MasterMix (Cordoba, Spain), 200 nM of the probe, 100 nM of each primer adding 100 ng of DNA in 5 µL to the reaction. A threshold cycle of 38 was set as cut-off for virus positivity.

### 2.4. RNA Extraction and EcPV2 Gene Expression Evaluation

Total RNA extraction of 5 sections (5 µm) from each FFPE sample was performed using RecoverAll Total Nucleic Acid Isolation Kit for FFPE (Invitrogen), according to the manufacturer’s instructions. *E6* and *E2* EcPV2 genes were tested for their expression using specific primers set and probe (Table 1) after the evaluation of RNA concentrations through NanoDrop 2000 (Thermo Fisher Scientific, Waltham, MA, USA) spectrophotometer. Reverse transcription (RT) step was performed using SuperScript IV VILO Master Mix (Invitrogen, ThermoFisher Scientific, Waltham, MA, USA) adding 500 ng of RNA. Then, 5 μL of 1:5 diluted complementary DNA (cDNA) was added to 20 μL PCR mixture at the final concentration of 1× master mix (iTaq Universal ProbsSupermix, Bio-Rad, Irvine, CA, USA), 200 nM of the probe, 100 nM of each primer combination. The reaction occurred in a CFX96 Real-Time System (BioRad, Hercules, CA, USA) with the following thermal profile: 95 °C for 10 min, then 39 cycles of 95 °C for 15 s and 60 °C for 60 s. RNA was used as the control to exclude possible contaminations by EcPV2 genomic DNA.

### 2.5. RT-qPCR for Host Gene Study

In this study, we evaluated gene modulation as a result of EcPV2 infection and cancer development. To this purpose, receptor activator of nuclear factor-kappa B ligand (*RANKL*), nuclear factor kappa-light-chain-enhancer of activated B cells (*NFKB*)*p50*, *NFKBp65*, interleukin (*IL*)*-6*, *IL17*, *IL23p19*, β-catenin (*BCATN1*), FOS like 1 (*FOSL1*), and lymphoid enhancer binding factor 1 (*LEF1*) were selected for gene expression evaluation. All the primers were designed through Primer3web tool v. 4.1.0 (https://primer3.ut.ee). In addition, 3 other genes, namely, *IL8*, *IL12p35*, and *IL12p40*, were tested using previously evaluated primer pairs [4,30]. *B2M* was chosen as reference gene to normalize relative gene expression evaluation. The primer set is reported in Table 2. As previously described [31,32] a Real-Time PCR amplification using SsoFast EvaGreen Supermix (BioRad, Hercules, CA, USA) was performed in a CFX96 Real-Time System with 5 μL of 1:5 diluted cDNA. Each sample was tested in triplicate and fluorescence data were collected at the end of the second step of each cycle. Relative expression was calculated through the ΔΔCq method.

### 2.6. Statistical Analysis

All continuous variables (gene expression values, Ki67, and mitotic count) were evaluated for normality distribution through the Shapiro–Wilk test. Non-parametric tests were used to test hypothesis. Mann–Whitney *U* test was performed to assess differences in gene expression among groups (T vs. C and *E6*+ vs. *E6*−). The Kruskal–Wallis *H* test was used to evaluate median differences between C, *E6*+, and *E6*− samples, applying a Mann–Whitney *U* test for pairwise comparison. Differential expression was performed in R (v. 4.0.3, R Foundation for Statistical Computing, Vienna, AT, Eur). https://www.r-project.org) environment while Spearman’s test (ρ) and descriptive statistic were performed in Microsoft Excel; other statistical tests were performed with IBM SPSS (v. 21, IBM Corp, Armonk, NY, USA). The *p-*value (P) threshold was set at 0.05 for the statistically significance.

## 3. Results

### 3.1. Case Selection, Histological Diagnosis of Tumors, Mitotic Count, and Ki67 Index

Twenty-three cases of equine genital epithelial neoplastic lesions were retrospectively selected from our archives; 1 case was diagnosed as papilloma, 2 cases as carcinoma in situ (CIS), whereas the 20 remaining cases were confirmed as invasive SCCs. The 10 selected control samples were histologically healthy. The median mitotic count in the T group was 41.56 (interquartile range-IQR = 22.41–60.51); Ki67 instead showed a median value of 30.90 (IQR = 20.84–43.51).

Spearman’s correlation (Table 3) showed no associations between mitotic count and Ki67, *RANKL*, *NFKB1*, *NFKBp65*, *IL23p19*, *LEF1*, *FOSL*, *BCATN1*, *IL6*, *IL8*, *IL17*, *IL12p35*, and *IL12p40*. No association with viral load or with *E6* expression was observed.

### 3.2. EcPV2 DNA Detection and EcPV2 Gene Expression

The B-2-microglobulin (*B2M*) gene was amplified in all samples, which were therefore considered suitable for the investigation of the viral gene *E6*, as shown in Table 4. Of the 23 cases, 21 (91%) cases were positive for EcPV2-*E6* viral DNA. The two negative samples were both SCCs. Among the positive samples, 6/21 showed a mean Cq between 34–38 (+), 2/21 between 29–33Cq (++), 4/21 of tested CCS showed a mean Cq 23–28 (+++), and 9/21 between 18–22Cq (++++). All samples collected from vulvar lesions showed high viral load (++++). *E6* gene was expressed (*E6*+) in 14 samples (60.8%), while 7 cases resulted in no expressing this viral gene (*E6*−). The expression of *E2* was detected in 8/23 samples (34.8%), indicating a lack of expression in six *E6+* samples (Table 4). All samples with high viral load (+++/++++) showed the expression of *E6* oncogene (*E6*+). Control samples were negative to EcPV2 DNA, as expected (Appendix A).

### 3.3. IL12 and IL23 Gene Expression

For *IL12p35*, *IL12p40*, and *IL23p19*, a statistically significant increase of the expression was observed in tumor samples (T) compared to healthy tissues (C). *IL12p35* medians were C_median =_ 0.23 and T_median =_ 18.66 (P = 7.8 × 10^−5^), *IL12p40* medians were C_median =_ 0.01 and T_median =_ 223.39 (P = 1.4 × 10^−7^), and *IL23p19* medians were C_median =_ 0.71 and T_median =_ 346.93 (P = 7.1 × 10^−8^) (Figure 1).

While no difference in expression was observed between E6+ and E6−, all tested genes were significantly modulated with respect to C, in both conditions (Appendix A). Moreover, a strong positive correlation between the expression of *IL23p19*-*IL12p40* (ρ = 0.775; P < 0.01) and *IL12p40-IL12p35* (ρ = 0.748; P < 0.01) (Table 3) was observed.

### 3.4. Inflammation

A statistically significant increase of the expression of all the tested genes was observed in T samples in comparison to C samples. In particular, *IL6* (C_median_ = 0.03, T_median_ = 4.92, P = 6.92 × 10^−6^), *IL8* (C_median_ = 0.004, T_median_ = 4.30, P = 2.76 × 10^−5^), *NFKB1* (C_median_ = 0.68, T_median_ = 4.62, P = 0.0003), *IL17A* (C_median_ = 0.23, T_median_ = 9.25, P = 6.92 × 10^−6^), and *NFKBp65* (C_median_ = 1.07, T_median_ = 3.94, P = 0.003) were upregulated in the T group (Figure 2). No significant differences in expression between *E6*+ and *E6−* were detected, although all tested genes had a different expression in both conditions when compared to the C group. Only for *IL6* and *NFKB1* were there different distributions of the median values among *E6*+ and *E6−*, but statistical significance was not reached (Appendix A). Spearman’s test showed a strong correlation between the expression of genes involved in inflammation (*NFKB1*, *NFKBp65*, *IL6*, and *IL8*) and *IL12* (*IL12p40* and *IL12p35*), as shown in Table 3. No correlation was observed with *IL17A*.

### 3.5. RANKL and Wnt Signaling Pathways

Concerning the Wnt signaling pathway and *RANKL*, all the tested genes showed a statistically significant increase of the expression in T samples compared to C (*RANKL* medians were C_median_ = 0.08 and T_median_ = 82.05, P = 0.0067; for *LEF1*C_median_ = 0.76 and T_median_ = 133.54, P = 1.43 × 10^−7^; *BCATN1*C_median_ = 1.48 and T_median_ = 25.93, P = 7.13 × 10^−8^; *FOSL1*C_median_ = 0.25 and T_median_ = 17.09, P = 0.0034) (Figure 3). Considering three groups (C, *E6*+, and *E6−*), no significantly differences in expression between *E6*+ and *E6−* were detected for all the tested genes, while the two groups remained statistically different with respect to controls, except for *FOSL1*, which was significantly upregulated in samples *E6*+ (*E6*+_median_ = 23.64, P = 0.0019) but not in *E6−*. Furthermore, Spearman’s test revealed a strong correlation between the expression of *RANKL* and genes involved in inflammation (*NFKB1*, *NFKBp65*, *IL6*, and *IL8*), *IL12* (*IL12p40* and *IL12p35*), IL23p19, and Wnt pathway (*BCATN1*, *FOSL1*, and *LEF1*) (Table 3). We did not observe correlation between *IL17A* and *RANKL* or Wnt pathway.

## 4. Discussion

The horse has been recently proposed as an animal model for human SCCs due to numerous histological similarities and the characteristic immune microenvironment of the genitalia tumors in the two species [4,33]. In both species, PV mucosal infections often induce cervical, oropharyngeal, penile, anal, vaginal, and vulvar cancer [34,35]; moreover, a case of equine gastric SCC has been linked with EcPV2 infection of the vulva [29]. The molecular mechanisms behind PV-induced tumor appears to be related to host local immune response by the tumor immune microenvironment (TIME) [36,37]. Indeed, tumors can occur decades after infection as a result of a defective immune response that allows persistent HPV infections [38,39] and consequent cancer development [36].

In our study, the presence of EcPV2, which is the most common equine PV in SCCs [40], was investigated by testing the samples for the DNA oncogene *E6* positivity. Out of 23 tumors, 21 (91%) resulted positive for EcPV2-*E6* DNA. The lack of *E6* positivity in the two SCC samples could be due to DNA fragmentation, a frequent condition in formalin-fixed and paraffin-embedded (FFPE) samples, although a specific primer pair able to produce an amplicon less than 100bp was used [29]. In humans, two pathways are involved in vulvar squamous cells carcinomas onset. The first is HPV-related, where usual-type vulvar intraepithelial neoplasia (uVIN) represents a well know SCC precursor. As shown by de Sanjosé et al. [41] in a recent study on over 2000 intraepithelial and invasive lesions, HPV-DNA was detectable in 86.7% of uVIN lesions. The other pathway leading to cancer is not related to HPV and is associated with chronic dermatoses, such as lichen sclerosus (differentiated VIN, dVIN), representing the most frequent precursor of SCC. For this reason, when the overall prevalence of HPV in vulvar invasive cancer was analyzed, only 30–43% were found to be HPV-DNA positive [41,42]. In penile carcinomas, the association with high-risk HPV infection is similar, and is reported to be present in about 50% of cases [43]. These data seem to be comparable to those previously obtained in horses [1] and are in line with our results that showed 60.8% of E6+ samples. Indeed, two different pathways seem to be associated with human vulvar and penile SCC, HPV-related and HPV-independent. The latter is often characterized by DNA mutation of TP53 and PTEN genes, allelic imbalance, microsatellite instability, loss of heterozygosity, or hypermethylation of tumor suppressor genes and genes acting for genome stability [44,45]. In HPV-related SCC, E6 and E7 viral oncogene products have a crucial role for carcinogenesis, interacting with host cell p53 and Rb proteins, leading to p53 dysfunction and Rb inactivation. Recently, these oncogenes have been found to target a plethora of other cellular factors, such as Myc and APOBEC3, and regulate the methylation status of cellular targets through the interaction with DNMT1 and other epigenetic modulators, which dictate the post-translational modifications on histones. This can lead to abolition of cell cycle arrest and hyperproliferation of tumor cells, evasion of apoptosis, DNA damage, and the suppression of the host cell immune response [46]. Of our 23 cases, over half (14) showed *E6* expression predominately associated with a high concentration of viral DNA. It is commonly postulated that some parts of the *E2* gene are usually disrupted or deleted from the HPV genome during the process of the viral DNA integration into the host genome. This gene regulates viral DNA transcription and replication; in cells where *E2* deletion or disruption occurs, most viral genes are found to be silenced, with the exception of *E6* and *E7* that, conversely, become overexpressed, causing tumor progression and a worse prognosis [44,46]. In our study, six of the *E6*+ samples lacked the expression of the *E2* gene and showed a tendency of increasing *E6* expression, indicating a possible viral DNA integration into the host genome (Table 4). This finding suggests that, also in horses, tumor progression might be associated with DNA integration and *E2* disruption, although this hypothesis must be confirmed in fresh/frozen samples to avoid the degradation of nucleic acids.

The expression of Ki67 was evaluated by immunohistochemistry (IHC) assay. Ki67 is a nuclear protein expressed in all proliferating vertebrate cells, and for this reason is widely used in routine tumor assessment, particularly for prognosis. In humans, IHC Ki67 staining is used in cervical and penile cancers for grade assessment, although in the latter it seems to have no prognostic value [47,48]. Although our results showed a relatively high Ki67 index (median = 30.90), no association with high viral content or E6 expression was detected, in agreement with what has been previously reported in horses. The reasons for these results can be various. For example, the poor homogeneity of the tumor with highly differentiated areas and low Ki67 levels alongside portions expressing a greater amount of this protein linked to their poor differentiation [40]. Moreover, long times from surgery to fixation or inadequate time of fixation could have a negative effect on Ki67 antibody binding [49]. Despite the fact that Ki67 staining is indicative of proliferation, it does not seem to be associated with any of the pathways investigated in this study.

Given the central role of the immune system in the pathogenesis of SCC, we investigated on the possible pathways relevant for the immune response regulation and also on their implication in tumor progression.

A signaling pathway recently associated with human genital cancer is RANK/RANKL, which seems to participate in immune cell proliferation and tumor cell survival [20,21] and in chronic inflammatory conditions [22]. The binding of the ligand RANKL to the RANK receptor activates a wide range of signaling cascades, including NF-kB [19], and promotes epithelial to mesenchymal transition (EMT) [24,50]. Notably, the current study reports, for the first time, the overexpression of the RANKL gene in egSCC (Figure 3), and this finding suggests a possible involvement of RANKL signal in egSCC EMT phenomena as described in humans [26]. This finding corroborates the idea that this pathway may represent an important pathway to consider for both prognostic and therapeutic reasons [51]. Considering that carcinoma cells in both in humans [26] and animals [52] express RANK mRNA/protein at various levels, often associated with a poor prognosis [23], the current findings may allow to consider *RANK* overexpression as a poor prognostic markers in the egSCC.

Moreover, RANKL upregulation has been reported in tumor-infiltrating T cells and in the locoregional lymph nodes [24]. RANK/RANKL pathway is intermingled with Wnt and IL17 signaling. Wnt signaling pathway has been considered one of the main contributors to the hallmarks of cancers due to the constitutive activation of target genes such as cMyc, CCND1, and VEGF [53,54]. Therefore, they are considered, in many carcinomas, to be a critical step for carcinogenesis [55,56,57,58].

It is known that Wnt canonical signaling deregulation is often a feature of oncogenic transformation in human cancer [8,12]. Interestingly, our results revealed, for the first time in egSCC, a significant upregulation of *RANKL*, together with *BCATN1*, *LEF1*, and *FOSL1* (Figure 3) genes, compared to the control group. These results might suggest a canonical Wnt pathway activation in the analyzed equine genital carcinomas, similarly to what is reported in both human penile squamous cell carcinomas [14] and cervical cancers [15]. Moreover, our results seem to be also in agreement with the results of a recent study on equine penile SCC that investigated the protein expression of a different downstream targets of the canonical Wnt pathway such as Cyclin D1, matrix metalloproteinase 7, c-myc, and fra1 [1].

*FOSL1* gene, encoding fra1 protein [59] and found to be significantly overexpressed in the present study, is a suggested target gene for the canonical Wnt pathway. It undergoes transcription when canonical Wnt signaling is activated [60,61]. Nevertheless, FOSL1 also takes part in the Ca2+ Wnt signaling, one of the two non-canonical Wnt pathways, together with the planar cell polarity pathway (Wnt/PCP). Interestingly, non-canonical Wnt signaling pathways have been reported to be implicated in the development of many cancers [62]. Our results suggest an activation of both canonical and non-canonical Wnt signaling pathway. This is similar to what has been reported for hepatocellular in humans, where canonical and non-canonical Wnt pathways are involved in tumor initiation and tumor progression, respectively [63]. Moreover, it is also noteworthy that the *E6*+ tumors displayed a significantly increased *FOSL1* expression compared to the *E6−* ones. In addition, E6 and E7 PV oncoproteins have already been suggested to promote epithelial to mesenchymal transition (EMT) in non-small cell lung cancer (NSCLC) cells [64] and that both canonical and non-canonical Wnt pathways have been associated with EMT [63]. In summary, it might be reasonable to speculate that the egSCC may undergo the EMT processes, as already preliminary suggested by Bonnet et al. [33].

In addition, it should be considered that Fra1, activated by the Wnt pathway, is also a target of the interleukin (IL) signaling cascade IL17RA [65]. This pathway is triggered by IL17A, a pro-inflammatory cytokine produced by a group of T helper cells known as Th17, in response to IL-23 stimulation. Our results revealed a significant upregulation of *IL17A* within the tumor group when compared to the controls explaining the potential role of *FOSL1* in the immune response [66].

Our results on *IL17A* are in agreement with a recent study conducted on HPV-related cancer [67], where Xue and colleagues demonstrated that Th17 cells and IL17A increased in cervical lesions when compared to controls. Cancer TIME is critical for the differentiation of naive CD4(+) T cells (Th0) into Th17 and Treg/Th17 balance. In particular, high levels of *IL23* gene expression have a pivotal role in this phenomenon [68] and on the consequent expression of *IL6* and *IL17A*. In this study, we demonstrated for the first time in egSCC, a high expression of *IL23* (*IL12p40* + *IL23p19*), *IL6*, and *IL17A*, suggesting a consistent presence of Th17 cells within the TIME. Moreover, *IL17A* seems to be involved in neutrophil accumulation, due to its ability to boost the expression of several chemokines, such as *IL8* [69]. In our previous study, we determined that egSCC tumors exhibit a strong immune infiltrate characterized by different inflammatory cells indicating a vivacious TIME in equine penile SCCs. We observed abundant TILs (tumor-infiltrating lymphocytes), TAMs (tumor-associated macrophages), and TANs (tumor-associated neutrophils) [4]. Our data confirm high gene expression of *IL8* with a significant difference between control and cancer; overall, these data suggest a Th17 activation and an imbalance between Th1/Th2 and Th17/Treg that can support tumor development, as reported in cervical cancer [7].

## 5. Conclusions

Our findings report for the first time RANKL gene expression in egCCS associated with the upregulation of BCATN1, FOSL1, and LEF1, suggesting the activation of Wnt signaling pathway that could be critical for carcinogenesis and tumor progression leading to speculate that the egCCS of the current study may undergo to EMT.

Our results describe a complex inflammatory TIME characterized by the activation of RANKL/RANK and IL17 pathways leading to the upregulation of proinflammatory cytokines, such as IL6 and IL8. Many of these molecules are involved in Th17 differentiation and Treg/Th17 imbalance.

However, further investigations on TIME in egSCCs might result to be helpful to identify new targets for immunotherapy strategies. Prospective studies are currently running to confirm our preliminary findings. These studies, in the era of the “One Health” approach, could confirm egSCCs as a model for the human disease.

## Figures and Tables

**Figure 1 biology-10-00244-f001:**
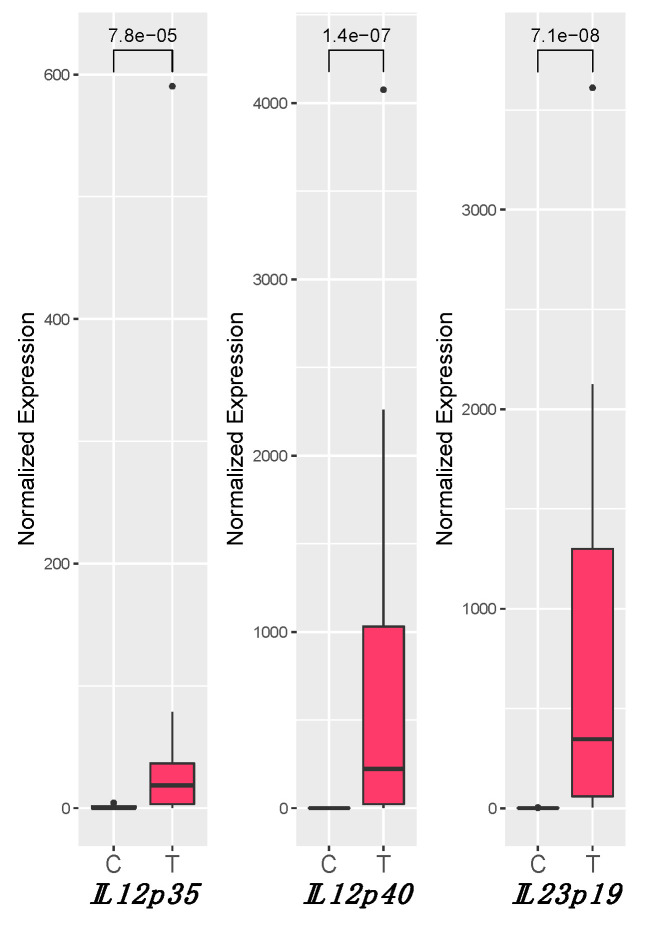
Gene expression levels of *IL12p35*, *IL12p40*, and *IL23p19*. The gene expression levels were normalized to that of the reference *B2M*. The Mann–Whitney *U*-test was used for the statistical analysis. The statistically significant (P < 0.05) values for each gene are reported on the horizontal bars in the upper part of the graph (“e” stands for exponential of ten). C: control group; T: tumor group.

**Figure 2 biology-10-00244-f002:**
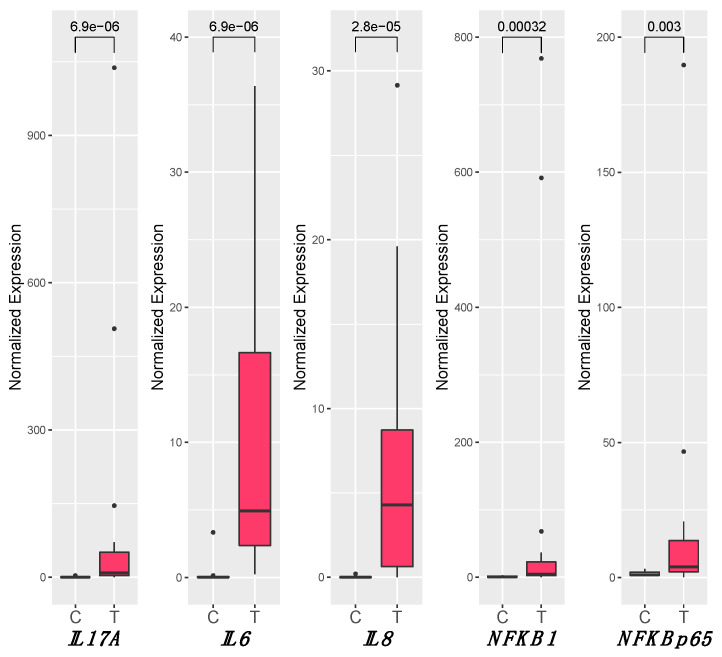
Gene expression levels of *IL6*, *IL8*, *NFKB1*, *IL17A*, and *NFKBp65*. The gene expression levels were normalized to that of the reference *B2M*, and the Mann–Whitney *U*-test was used to assess the statistical significance (P < 0.05) of median differences. The statistically significant values among each comparison are shown above the horizontal bar on the upper part of the graph (“e” stands for exponential of ten). C: control group, T: tumor group.

**Figure 3 biology-10-00244-f003:**
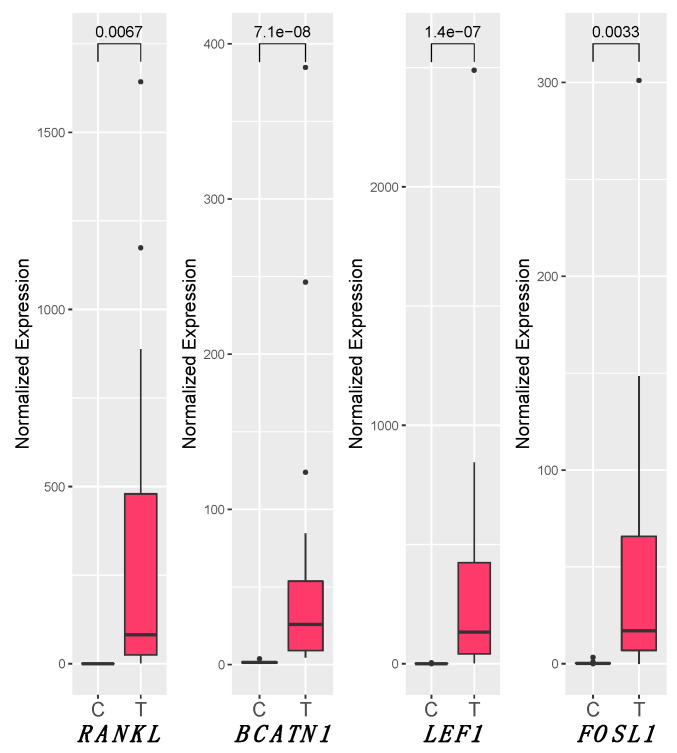
Box plot of *RANKL*, *LEF1*, *BCATN1*, and *FOSL1* gene expression. The statistical significance (P < 0.05) of median differences was assessed through Mann–Whitney *U* test after normalizing expression levels with those of the reference *B2M*. The statistically significant values are shown above the horizontal bar above the horizontal bar on the upper part of the graph (“e” stands for exponential of ten). C: control group; E6+: samples expressing E6 gene; E6−: samples not expressing E6.

**Table 1 biology-10-00244-t001:** Primer set and probes used to evaluate equine papillomavirus 2 (EcPV2) DNA detection and gene expression.

Gene	Sequences	Amplicon Length	Accession Number
EcPV2-*E2*	F-5′-AAAAGGGAGGGTACGTTGTC-3′ R-5′-CCTGGTAGTAGACATGCTGC-3′	90	NC_012123.1:2767–4017
p-EcPV2-*E2*	FAM- GCCAAGACAGCCACGACGCCAT-TAMRA	22	
EcPV2-*E6*	F-5′-CGTTGGCCTTCTTTGCATCT-3′ R-5′-AGGTTCAGGTCTGCTGTGTT-3′	81	NC_012123.1:5–622
p-EcPV2-*E6*	FAM- CCGTGTGGCTATGCTGATGACATTTGG -TAMRA	27	
*B2M* DNA detection	F-5′-CTGATGTTCTCCAGGTGTTCC-3′ R-5′-TCAATCTCAGGCGGATGGAA-3′	114	NM_001082502.3 chr1:145961271-145961384
*B2M* cDNA expression	F-5′-GGCTACTCTCCCTGACTGG-3′ R-5′- TCAATCTCAGGCGGATGGAA-3′	136	NM_001082502.3 chr1:145961271-145964672
p-*B2M*	FAM-ACTCACGTCACCCAGCAGAGA-TAMRA	21	NM_001082502.3

**Table 2 biology-10-00244-t002:** Primer set for the host gene expression evaluation.

Gene	Primer Pairs Sequences	Amplicon Length	Accession Number
*B2M*	F-5′-GGCTACTCTCCCTGACTGG-3′ R-5′-TCAATCTCAGGCGGATGGAA-3′	136	NM_001082502.3
*RANKL*	F-5′-AGCCTGACACTCAACCTTTTG-3′ R-5′-CCAGGAAGACAGACTCACTTTG-3′	86	XM_014732051.2
*NFKBp50*	F-5′-CCAGCTTTTGGTAGATGTGCTG-3′ R-5′-TCGTCTTCTGCCATTCTGGA-3′	102	XM_023637631.1
*NFKBp65*	F-5′-GAGCCCATGGAGTTCCAGTA-3′ R-5′-AGGTCTCATATGTCCTTTTGCGT-3′	82	XM_023654462.1
*IL8*	F-5′-CTGGCTGTGGCTCTCTTG-3′ R-5′-CAGTTTGGGATTGAAAGGTTTG-3′	133	NM_001083951.2
*IL17*	F-5′-ACAACCGCTCCACCTCCC-3′ R-5′-CCTTCGGCATTGACACAGC-3′	112	NM_001143792.1
*IL23p19*	F-5′-CTGTACGCTGGCCTGGAG-3′ R-5′-GTGGATCCTTTGCAAGCAGG-3′	167	NM_001082522.2
*IL6*	F-5′-TCAAGGGTGAAAAGGAAAACATC-3′ R-5′-GGTGGTTACTTCTGGATTCTTC-3′	98	NM_001082496.2
*IL12p35*	F-5′-CTGAGGACCGTCAGCAACAC-3′ R-5′-GTTCGGGGCGAGTTCCAG-3′	147	NM_001082511.2
*IL12p40*	F-5′-GATCGTGGTGGATGCTGTTC-3′ R-5′-TCCACCTGCCGAGAATTCTT-3′	132	NM_001082516.1
*BCATN1*	F-5′-CCTCTTCAGAACGGAGCCAA-3′ R-5′-CTGGCGATATCCAAGGGGTT-3′	91	XM_023619816.1
*FOSL1*	F-5′-TACCGAGACTTCGGGGAAC-3′ R-5′-GCGTTGATACTTGGCACGAG-3′	116	XM_001494776.4
*LEF1*	F-5′-GCCAGACAAGCACAAACCTC-3′ R-5′-GGGTCCCTTGCTGTAGAGG-3′	102	XM_023636760.1

**Table 3 biology-10-00244-t003:** Correlation matrix (Spearman rank correlation coefficient, ρ).

	Mitoses	Ki67	*BCATN1*	*FOSL1*	*IL12p35*	*IL12p40*	*IL17A*	*IL23p19*	*IL6*	*IL8*	*LEF1*	*NFKB1*	*NKFBp65*	RANKL
Mitoses	1.000	−0.074	0.117	−0.071	−0.145	0.160	−0.162	0.154	0.026	−0.161	−0.010	0.133	−0.004	0.153
Ki67		1.000	−0.370	−0.518	−0.389	−0.141	−0.071	−0.101	−0.155	−0.156	−0.171	−0.405	−0.402	−0.272
*BCATN1*			1.000	0.791 **	0.864 ^**^	0.657 **	0.161	0.800 **	0.888 **	0.371	0.890 **	0.425 *	0.486 *	0.773 **
*FOSL1*				1.000	0.758 ^**^	0.282	−0.011	0.412	0.698 **	0.469	0.610 *	0.352	0.533	0.632 *
*IL12p35*					1.000	0.742 **	0.235	0.748 **	0.893 **	0.451 *	0.868 **	0.520 *	0.381	0.707 **
*IL12p40*						1.000	0.432	0.775 **	0.809 **	0.162	0.739 **	0.481 *	0.184	0.634 **
*IL17A*							1.000	0.227	0.339	0.033	0.210	0.143	0.157	0.156
*IL23p19*								1.000	0.881 **	0.191	0.910 **	0.458 *	0.296	0.850 **
*IL6*									1.000	0.429	0.922 **	0.398	0.381	0.792 **
*IL8*										1.000	0.247	0.162	0.356	0.248
*LEF1*											1.000	0.454 *	0.267	0.856 **
*NFKB1*												1.000	0.227	0.515 *
*NFKBp65*													1.000	0.205

*: P < 0.05; **: P < 0.01.

**Table 4 biology-10-00244-t004:** Histological diagnosis: SCC: squamous cell carcinoma; CIS: carcinoma in situ; P: papilloma. RT-PCR data for beta-2-microglobulin (B2M) are expressed as + (amplified) or – (not amplified); indication of the viral amount was given through the Cq, at which point the positivity for E6 gene was detected: - (>38 Cq), + (34–38 Cq), ++ (29–33 Cq), +++ (23–28 Cq), and ++++ (18–22Cq). RT-qPCR data are expressed as mean Cq ± 1-standard deviation of three replicates. All samples were classified as E6+ when showing E6 expression (Cq < 48) and into E6− when E6 gene was not expressed (Cq > 48). ND indicates no amplification.

Case ID	Histological Diagnosis	Sex	DNA		cDNA		
			*B2M*	*E6*	*E6*	*E2*	E6 Expressing samples
1	SCC	M	+	++	30.4 ± 0.2	>48	*E6*+
2	SCC	M	+	+	>48	>48	*E6*−
3	SCC	M	+	+++	32.8 ± 0.3	>48	*E6*+
4	CIS	M	+	++++	36.2 ± 0.6	44.2 ± 2.4	*E6*+
5	SCC	F	+	++++	31.5 ± 0.7	34.1 ± 0.4	*E6*+
6	P	M	+	+	>48	>48	*E6−*
7	SCC	M	+	+	>48	>48	*E6−*
8	SCC	M	+	+++	33.5 ± 0.4	>48	*E6*+
9	SCC	M	+	++++	32.7 ± 0.9	36.1 ± 0.6	*E6*+
10	SCC	M	+	+++	33.5 ± 0.3	39.3 ± 1.1	*E6*+
11	SCC	M	+	+	>48	>48	*E6−*
12	SCC	M	+	++++	33.7 ± 0.4	>48	*E6*+
13	SCC	M	+	+	>48	>48	*E6−*
14	SCC	M	+	+	>48	>48	*E6−*
15	SCC	M	+	++++	33.9 ± 1.9	38.5 ± 0.6	*E6*+
16	SCC	F	+	++++	31.9 ± 0.4	>48	*E6*+
17	SCC	M	+	+++	35.3 ± 1.5	39.2 ± 0.9	*E6*+
18	SCC	M	+	++	>48	>48	*E6−*
19	SCC	F	+	++++	32.1 ± 1.1	35.6 ± 0.7	*E6*+
20	SCC	M	+	−	ND	ND	*E6−*
21	CIS	M	+	++++	30.4 ± 0.6	>48	*E6*+
22	SCC	M	+	++++	32.3 ± 0.1	41.9 ± 2.6	*E6*+
23	SCC	M	+	−	ND	ND	*E6−*

## Data Availability

Not applicable.

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
