# Peer review of "Equine Genital Squamous Cell Carcinoma Associated with EcPV2 Infection: RANKL Pathway Correlated to Inflammation and Wnt Signaling Activation"

_biology, 2021, doi:10.3390/biology10030244_

Round 1

Reviewer 1 Report

The work by Mecocci et al. aimed at clarifying the molecular mechanisms behind eqSSC, investigating RANKL, Wnt and IL17 signaling pathways. EcPV2 infection is often associated with eqSSC and in our study viral presence was confirmed in 21 out of 23 cases. Results described an inflammatory microenvironment characterized by increased expression of several cytokines (IL6, IL8, IL12p35, IL12p40, IL17), the activation of RANKL/RANK and IL17A signaling pathways. Increased expression of BCATN1, FOSL1 and LEF1 suggested activation of both canonical and non-canonical Wnt signaling pathways.

The work is well written and easy to read. Results are clearly presented and thoughtfully discussed. Methods are described in details. Overall I consider the work of Mecocci et al. defeat of major weakness. There only minor points I would suggest to address before considering the paper suitable for publication.

Line 36/37 and 443. Although I agree that this study might contribute to identify new target for immunotherapy, I don’t think it will contribute to identify prognostic factors, because you did not correlate your results with the level of malignancy or survival rate. Please slightly re-phrase this sentence.

Line 41. RANKL is mention for the first time here, not in line 46/47.

Line 45. You detected only viral genome, please rephrase here, table 1 title (line 161), and in paragraph 3.2.

Line 219-220. Did you assessed mitotic count also in the control group?

Line 224. Correlation analysis of what? ‘Correlation matrix based on…’

Why didn’t you assess also IL-10 or TGF-b expression, which are cytokines modulated in other cancers?

Author Response

Line 36/37 and 443. Although I agree that this study might contribute to identify new target for immunotherapy, I don’t think it will contribute to identify prognostic factors, because you did not correlate your results with the level of malignancy or survival rate. Please slightly re-phrase this sentence.

We thank the reviewer for this observation and we modified the sentences according to the suggestion.

Line 41. RANKL is mention for the first time here, not in line 46/47.

Error fixed.

Line 45. You detected only viral genome, please rephrase here, table 1 title (line 161), and in paragraph 3.2.

Sorry for the inaccuracy, sentences have been changed accordingly.

Line 219-220. Did you assessed mitotic count also in the control group?

Mitotic count is a histological value that is assessed in tumors and, in some cases, associated with prognosis; so, we did not evaluate it on normal mucosa.

Line 224. Correlation analysis of what? ‘Correlation matrix based on…’

The title of the table has been changed.

Why didn’t you assess also IL-10 or TGF-b expression, which are cytokines modulated in other cancers?

Dear reviewer, IL10 and TGFB gene expression were evaluated on FFPE samples in the context of our previous study about immune infiltrate in Equine SCC. In that circumstance, only a small percentage of samples showed a detectable IL10 and TGFB expression; consequently, we decided to not assess further these parameters in the current study.

Reviewer 2 Report

This manuscript by Mecocci et al describes some molecular characteristics of equine genital SCC. The authors conclude that egSCC harbors similar molecular features to the human disease and also for the first time associate upregulated RANKL expression with upregulated Wnt signaling components in egSCC. 

Overall, the paper is well-written and the data are presented in a sound manner, and the conclusions do not over-reach the actual data presented. This straight-forward molecular characterization of egSCC provides value to its particular field. 

The introduction and discussion sections are especially thorough and informative. 

No revisions are recommended by this reviewer. 

Author Response

We thank the reviewer for the positive comments.

This manuscript is a resubmission of an earlier submission. The following is a list of the peer review reports and author responses from that submission.